# Comparison of low-salt preference trends and regional variations between patients with major non-communicable diseases and the general population

**Eun Young Choi[1], Young-Kwon Park[2], Minsu Ock[2,3] ***

**1** College of Nursing, Sungshin Women's University, Seoul, Republic of Korea, **2** Prevention and Management Center, Ulsan Regional Cardiocerebrovascular Disease Center, Ulsan University Hospital, Ulsan, Republic of Korea, **3** Department of Preventive Medicine, Ulsan University Hospital, University of Ulsan College of Medicine, Ulsan, Republic of Korea

\* ohohoms@naver.com

**Data Availability Statement:** All relevant data are within the paper and its Supporting Information files.

## Abstract

### Background

We compared low-salt preference between patients with major chronic diseases, also known as non-communicable diseases (hereinafter referred to as "'major NCD patients'"), and the general population.

### Methods

We used data extracted from the Korea Community Health Survey Community during the period of 2008–2019. We evaluated the low-salt preference of 13 major NCD patients by year and region to analyse recent changes in low-salt preference trends, using joinpoint regression.

### Results

A greater majority of major NCD patients had a higher low-salt preference than general population; however, the overall trend was not significantly significant. The low-salt preference rate (Type III) was highest among patients with diabetes (15.6%), followed by hypertension (14.1%) and dyslipidaemia (13.4%), with the general population displaying the lowest rate (11.3%). The rates of adherence to a low-salt diet and fried food without soy sauce gradually increased until 2013 and then declined. The rate of adherence to not adding salt and soy sauce at the table gradually increased and maintained a high rate of adherence until 2019, except for patients with some diseases. Regional variations in low-salt preference tended to be greater in patients with major NCDs than general population.

### Conclusion

There is a need to improve the low-salt diet behaviour of not only major NCD patients, but also the general population. Various low-salt diet programs need to be promoted, such as

**Funding:** This work was supported by the Research Program funded by the Korea Disease Control and Prevention Agency. The funders had no role in study design, data collection and analysis, decision to publish, or preparation of the manuscript.

**Competing interests:** The authors have declared that no competing interests exist.

education on a low-salt diet, certification for low-salt restaurants, and sodium tax. Moreover, it is necessary to continuously monitor the low-salt preferences of major NCD patients.

## Introduction

Although it is well known that improvement of health behaviour is necessary for the management of non-communicable diseases (NCDs) such as hypertension, diabetes, and dyslipidaemia [1–3], it is challenging to modify health behaviour [4, 5]. However, there is no lack of opportunities for health behaviour improvement, and the theory of health behaviour recommends taking advantage of such cues to action. In particular, the situation of being diagnosed with an NCD and treated is recognised as a good opportunity for becoming aware of the health crisis and thus preventing the exacerbation of the disease and the development of comorbid diseases [6, 7] (i.e., what is known as the 'wake-up call' or a teachable moment).

Despite various efforts to improve NCD patients' health behaviour [8, 9], there is currently insufficient evidence to suggest that their health behaviours are significantly better than those of the general population [10]. In a study that followed the trends of three health behaviours (current smoking, alcohol consumption, and walking) in major NCD patients (hypertension, diabetes, and dyslipidaemia) and compared them with those of the general population by analysing the relevant Korea Community Health Survey data covering the period of 2008–2017, major NCD patients showed only marginally more health behaviour improvement compared with the general population, partially showing a lower level of improvement than the general population [10]. Regarding nutrition-related behaviours, some studies reported that patients improved their dietary behaviour through education or counselling after diagnosis of NCDs [11, 12]. In other studies, patients have complained of difficulties in correcting and continuing to fulfil their dietary behaviours that were formed over a long period of time [13–18].

The present study focused on major NCD patients' nutritional behaviours as a follow-up analysis of prior research, in which smoking, drinking, and physical activity (i.e., three typical health behaviours) were analysed; however, no analysis of nutrition-related health behaviours was undertaken. The World Health Organization (WHO) emphasises the importance of reducing unhealthy food and drink consumption, along with reducing tobacco use, harmful use of alcohol, and physical inactivity; these are the 'best buys' of the strategies to reduce the burden of NCDs [19]. Therefore, this study compared low-salt preference, which is a typical nutrition-related health behaviour, between major NCD patients and the general population, using the analysis methodology adopted in prior research.

## Materials and methods

### Data source

In this study, we used data extracted from the Korea Community Health Survey (KCHS) during the period of 2008–2019. The KCHS is a nationwide survey that was first conducted in 2008 by the Korea Centers for Disease and Control and Prevention to evaluate the health status of community residents and to establish evidence-based health-related statistics [20]. This survey was conducted each year from August to October at 255 community health centres across the country to evaluate anthropometrical parameters, smoking status, alcohol use, physical activity, health behaviours (e.g., diet and use of healthcare services), and quality of life in Korean adults (≥19 years of age) [21]. The sample size of KCHS is about 900 people in each community health center, for a total of about 220,000 people.

## Participants

Thirteen major types of NCD patients (hypertension, diabetes, dyslipidaemia, stroke, myocardial infarction, angina, arthritis, osteoporosis, asthma, allergic rhinitis, atopic dermatitis, cataract, and depression) were defined as participants who responded clinically diagnosed from physicians in the KCHS. Depression was subdivided into patients who had been diagnosed by a physician and those who had experienced symptoms. The general population was defined as all the participants in the Korea Community Health Survey.

## Low-salt preference

Low-salt preference was measured using four items: the percentage of respondents who answered that they 1) usually eat less salty food, 2) do not add salt or soy sauce at the table, 3) do not dip fried food in soy sauce, and 4) stick to a low-salt diet in the three previous items, that is, they have a low-salt preference (Type III).

## Data analysis

Descriptive statistical analysis of data was conducted to determine the low-salt preferences of NCD patients and the general population according to the year (2008–2019) and region (municipalities and provinces). A Chi-square test was performed to compare the low-salt preferences of patients with major NCDs and the general population. The 2018 Korea Community Health Survey data were excluded from the analysis because there were no survey items on low-salt preferences. Since the 2018/2019 data for the prevalence of dyslipidaemia were missing, we used the 2017 data. Joinpoint regression was applied to detect significant changes in annual low-salt preference [22], with a maximum of one joinpoint allowed. The annual percentage change (APC) (i.e., the average annual percentage change [AAPC] in the case of one joinpoint) was computed using joinpoint analysis, and each p-value is presented.

Analysis data were tabulated using Microsoft Excel 2014 (Microsoft Corporation, Seattle, WA). Stata/SE13.1 (StataCorp, Texas, TX) were used for descriptive statistics and Joinpoint Regression Program (version 4.6.0; US National Cancer Institute, Bethesda, MD, USA) were used for joinpoint regression.

## Ethical considerations

Since this study used a publically available secondary data source (Korea Community Health Survey, Available from: https://chs.kdca.go.kr/chs/main.do), we did not seek approval from the institutional review board. We also did not have to ask for the consent of the participants.

## Results

As of 2017, low-salt preference among major NCD patients was generally higher than that of the general population, however, overall statistical significance was not reached (Fig 1). The percentage of respondents who answered that they usually eat less salty food was 23.6% in the general population and 25.2%, 27.7%, and 24.4% in patients with hypertension, diabetes, and dyslipidaemia, respectively. The percentage of respondents who answered that they did not add salt or soy sauce at the table was slightly higher in patients with hypertension and dyslipidaemia (72.2% and 74.0%, respectively) than in the general population (71.9%), though the percentage was slightly lower (71.5%) in patients with diabetes. The percentage of respondents who answered that they did not dip fried food in soy sauce was slightly higher in major NCD patients than in the general population (36.0%). The percentage of respondents who indicated that they preferred a low-salt diet for all three items was highest in patients with diabetes

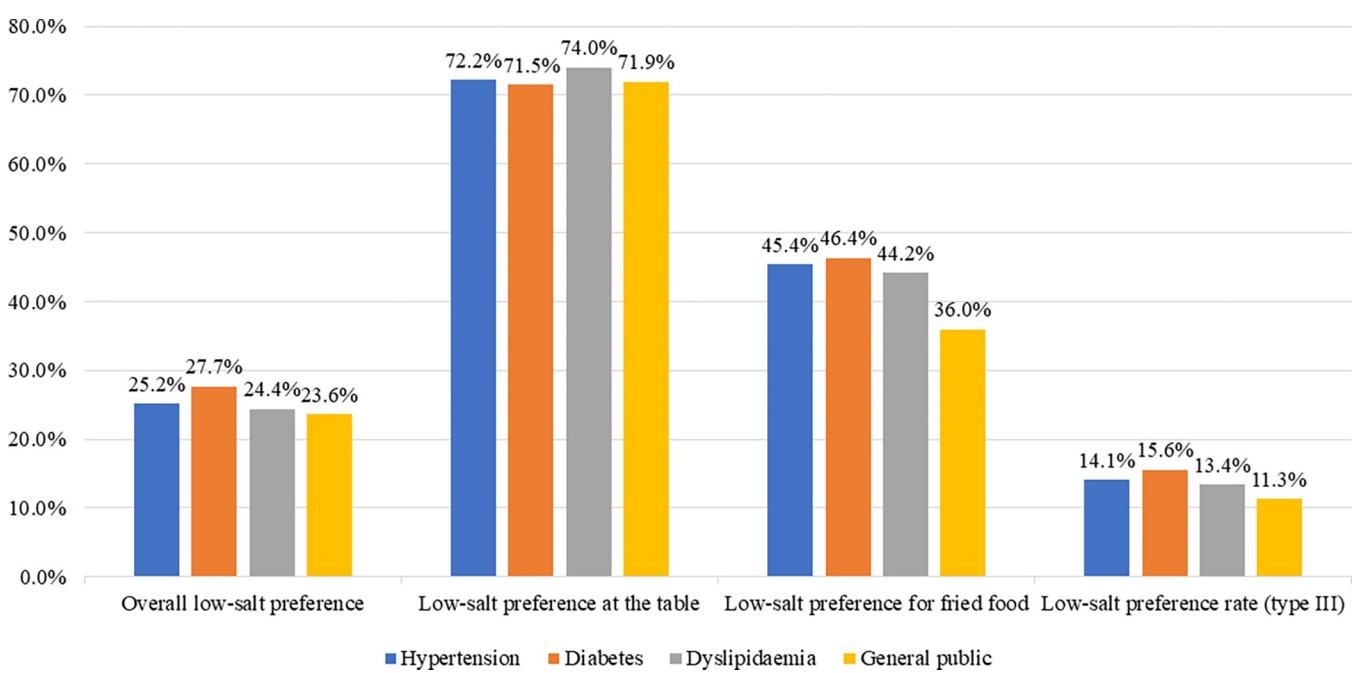

**Fig 1. Comparison of low-salt preference of patients with major non-communicable diseases (as of 2017).**

(15.6%), followed by patients with hypertension (14.1%), dyslipidaemia (13.4%), and the general population (11.3%).

## Overall low-salt preference

The overall low-salt preference rate (usually eat less salty food) showed an upward trend until 2013 and declined thereafter (Fig 2). In the general population, it peaked in 2013 at 24.9%

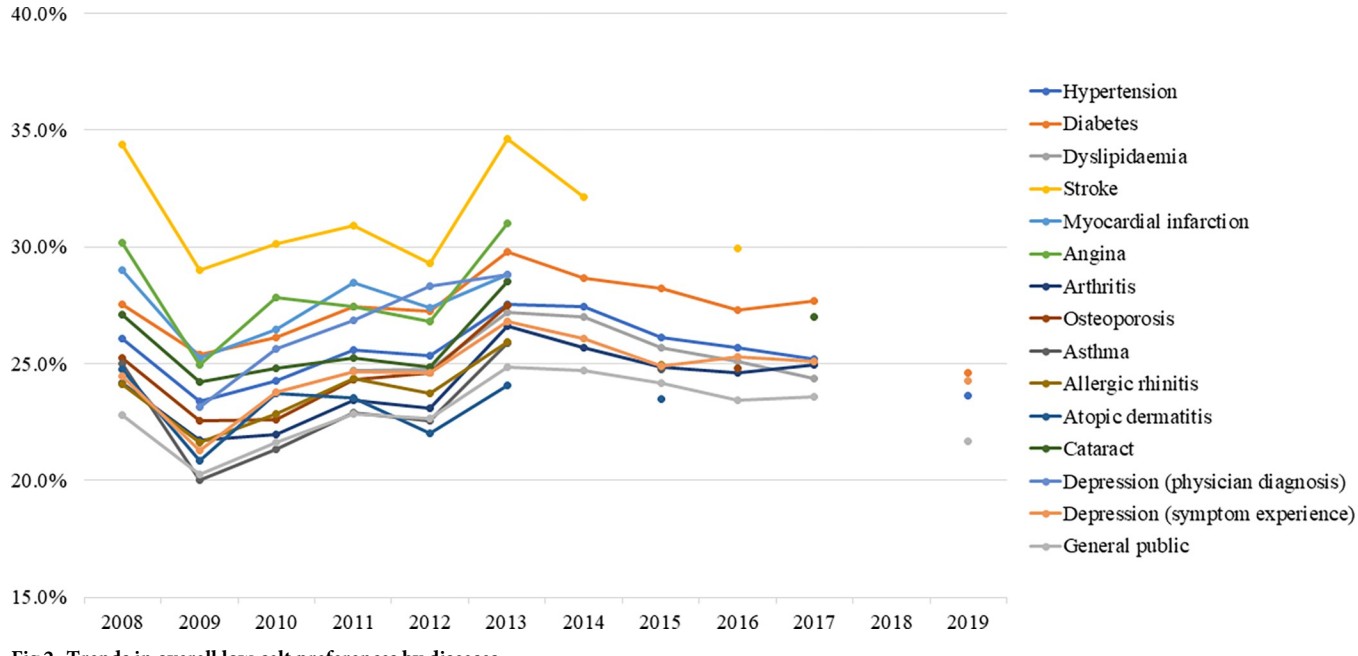

**Fig 2. Trends in overall low-salt preferences by diseases.**

(APC: 2.654, p = 0.063) and decreased to 21.6% by 2019 (APC: -2.347, p = 0.195). In patients with hypertension, it peaked in 2013 at 27.6% (APC: 1.857, p = 0.136) and decreased to 21.6% by 2019 (APC: -2.649, p = 0.126). In patients with diabetes, it peaked in 2013 at 29.8% (APC: 1.859, p = 0.122) and decreased to 23.6% by 2019 (APC: -2.849, p = 0.093). Lastly, in patients with dyslipidaemia, it peaked in 2013 at 27.2% (APC: 5.690, p = 0.048) and decreased to 24.4% by 2017 (APC: -2.528, p = 0.023).

## Low-salt preference at the table

The rate of low-salt preference at the table (not adding salt and soy sauce at the table) gradually increased and tended to remain elevated; in patients with some diseases, the rate slightly decreased by 2019 (Fig 3). In the general population, it peaked in 2013 at 74.4% (APC: 11.788, p-value: <0.001) and decreased to 68.4% by 2019 (APC: -0.964, p = 0.010). In patients with hypertension, it peaked in 2013 at 73.5% (APC: 11.401, p = 0.001) and decreased to 68.1% by 2019 (APC: -0.523, p = 0.225). In patients with diabetes, it peaked in 2013 at 72.7% (APC: 10.914, p <0.001) and then gradually decreased to 68.3% by 2019 (APC: -0.567, p = 0.113). In patients with dyslipidaemia, there was a slight decrease from 75.9% in 2011 to 74.0% in 2017 (APC: -0.390, p-value: 0.109).

## Low-salt preference for fried food

The rate of not dipping fried food in soy sauce increased until 2013 and then decreased (Fig 4). No significant changes were observed in the general population between 2008 (36.8%) and 2019 (36.4%), and there were fluctuations in some years (APC: 0.214, p = 0.693). In patients with hypertension, the rate peaked in 2013 at 50.3% (APC: 2.993, p = 0.023) and then decreased to 44.9% by 2019 (APC: -1.585, p = 0.092). In patients with diabetes, it peaked in 2013 at 50.3% (APC: 2.494, p = 0.035) and then decreased to 45.2% by 2019 (APC: -1.634, p = 0.070). Lastly, in patients with dyslipidaemia, it peaked in 2013 (47.9%) and then decreased to 44.2% by 2017 (APC: -0.152, p = 0.847).

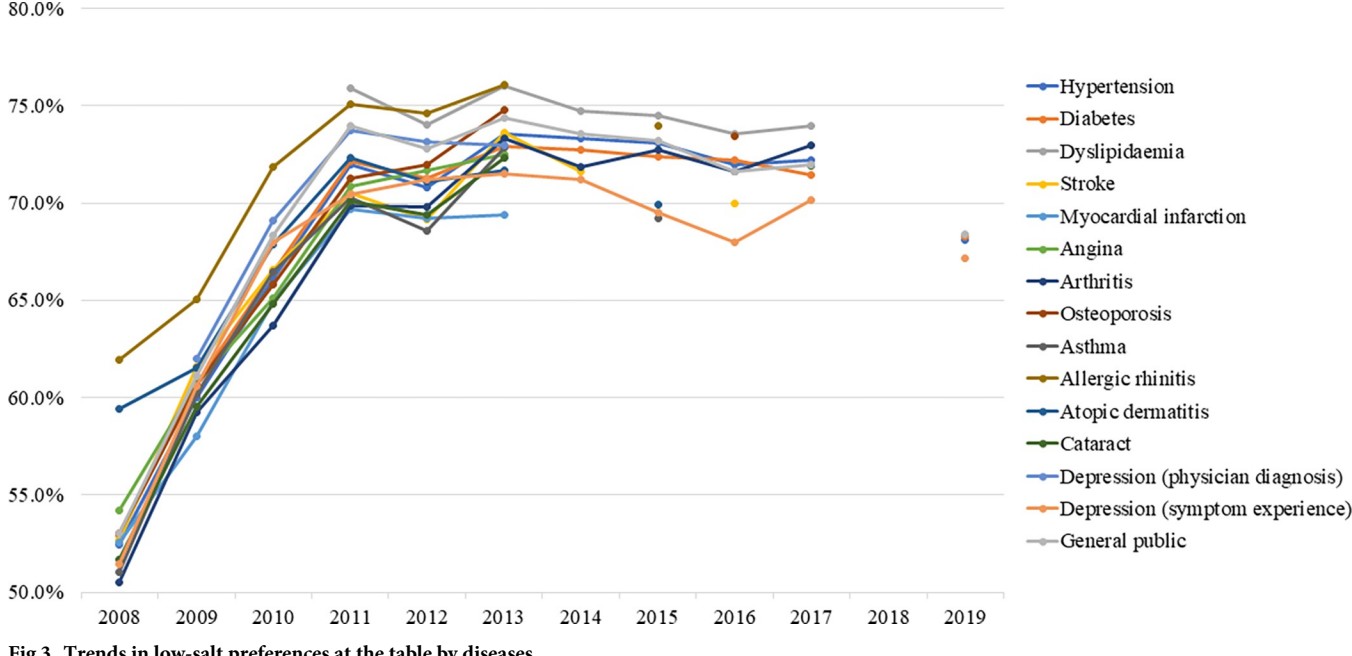

**Fig 3. Trends in low-salt preferences at the table by diseases.**

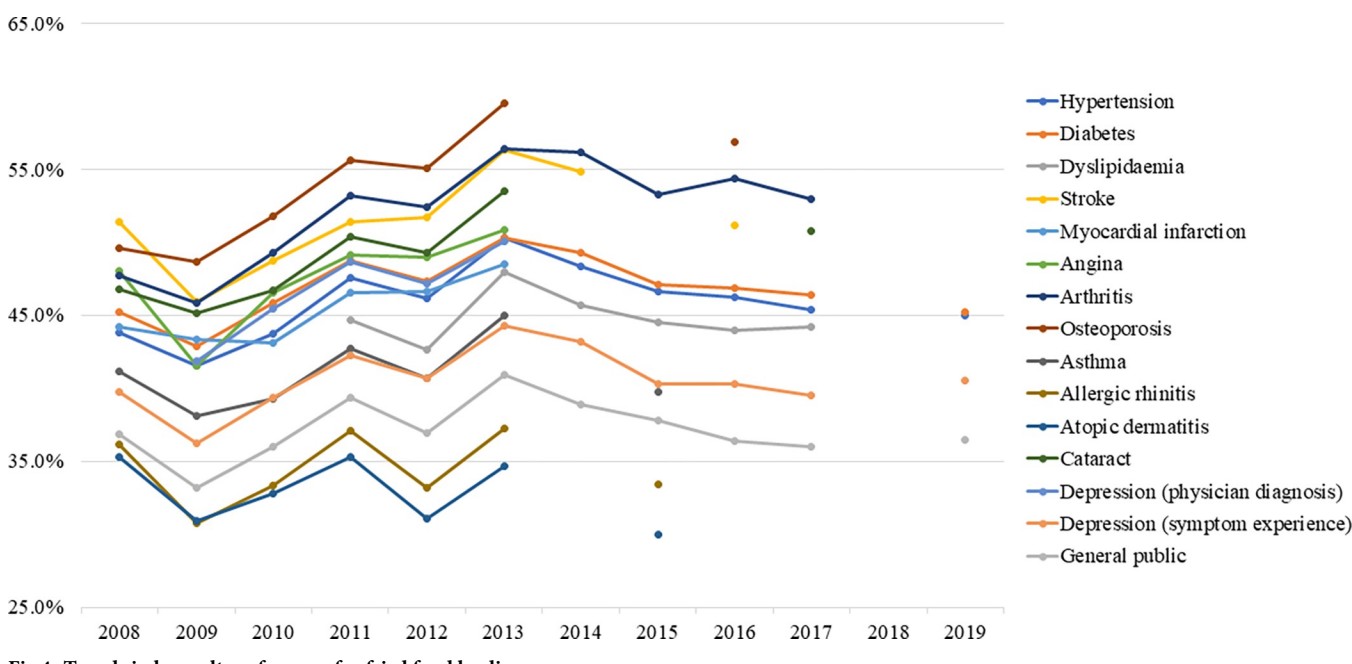

**Fig 4. Trends in low-salt preferences for fried food by diseases.**

## Low-salt preference rate (type III)

Low-salt preference (type III) increased until 2013 and decreased thereafter (Fig 5). In the general population, it peaked in 2013 at 13.2% (APC: 7.528, p = 0.009) and then decreased to 10.4% by 2019 (APC: -3.422, p = 0.71). In patients with hypertension, it peaked in 2013 at 16.3% (APC: 4.621, p = 0.011) and decreased to 12.8% by 2019 (APC: -4.638, p = 0.037). In patients with diabetes, the low-salt preference rate (type III) peaked in 2013 at 18.1% (APC:

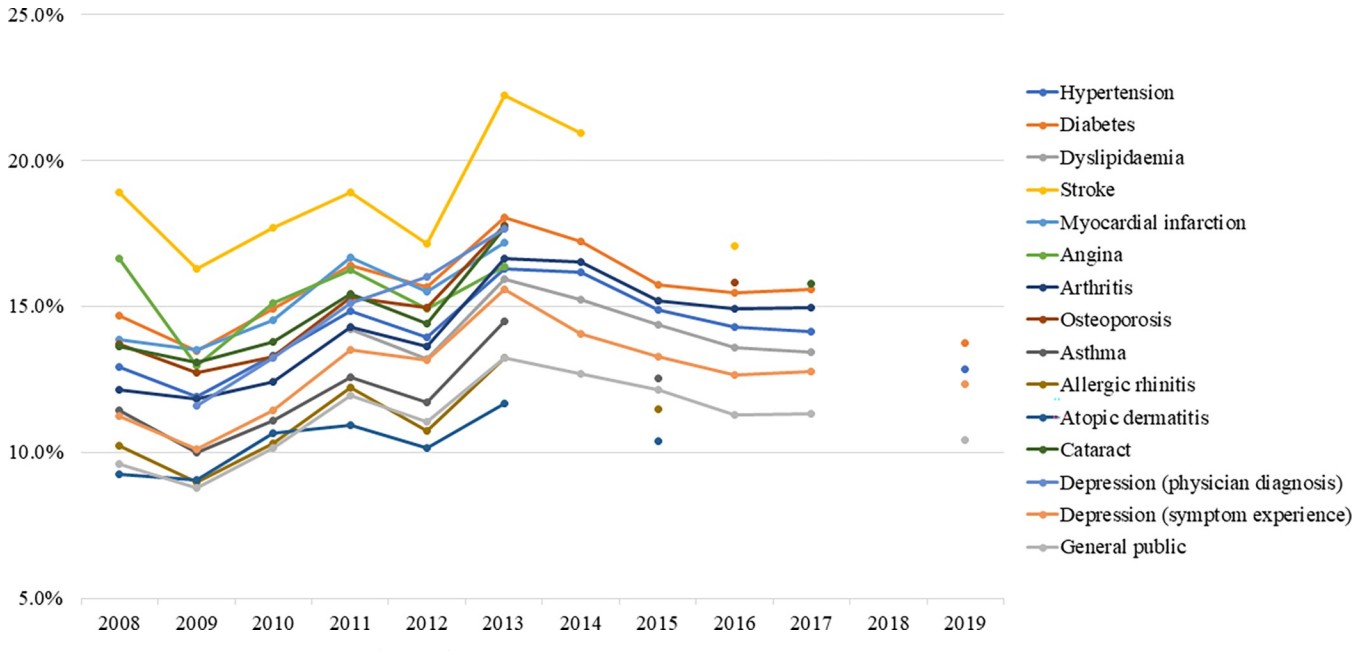

**Fig 5. Trends in low-salt preferences rate (type III) by diseases.**

4.761, p = 0.024) and decreased to 13.7% by 2019 (APC: -3.634, p = 0.027). Although no consistent rate was observed in patients with dyslipidaemia, a slight decrease was observed from 14.2% in 2011 to 13.4% in 2017 (APC: -0.759, p = 0.603).

## Regional variations

Greater regional variations in low-salt preference were observed in major NCD patients compared to the general population (S1 File). In 2019, Seoul and Jeonnam had the highest and lowest percentages of respondents who answered that they usually ate bland or less salty food (24.3% and 18.3%, respectively; standard deviation [SD]: 1.7%). For patients with diabetes, Jeju and Gyeongnam were the regions with the highest and lowest percentages (30.1% and 20.8%, respectively; SD: 2.4%). Gwangju and Daejeon were the regions with the highest and lowest percentages of the general population who answered that they did not add salt or soy sauce at the table in the same year (77.0% and 54.1%, respectively; SD: 5.7%). Similar regional variations were observed in patients with hypertension: Gwangju, 81.1%; Daejeon, 58.2%; SD, 5.8%. Jeonnam and Daegu were the regions with the highest and lowest percentages of the general population who answered that they did not dip fried food in soy sauce, at 55.8% and 23.6%, respectively (SD: 8.0%). For patients with diabetes, Geonnam and Sejong were the regions with the highest and lowest percentages (65.0% and 20.3%, respectively; SD: 11.4%). With regard to the low-salt preference rate (type III), Seoul and Daegu were the regions with the highest and lowest percentages, at 12.3% and 7.6%, respectively (SD: 1.4%). In patients with diabetes, the regions with the highest and lowest percentages were Jeonnam (17.0%) and Gangwon (8.6%), respectively (SD: 2.5%).

## Discussion

In this study, we investigated the low-salt preferences of major NCD patients (e.g., hypertension, diabetes, and dyslipidaemia) from various angles and compared them with those of the general population based on the relevant datasets extracted from the Korea Community Health Survey. Data analyses revealed that major NCD patients had a higher low-salt preference compared with the general population; however, the difference was not statistically significant. Despite an overall upward trend in low-salt preferences, the low-salt preference rates of patients with different major NCDs and the general population was gradually decreasing after peaking mainly in 2013; thus, the low-salt preference rates leave much room for improvement. In addition, we found greater regional variations in low-salt preferences among major NCD patients compared to the general population. This finding highlights the need to prepare a region-specific strategic approach to improve low-salt preferences.

Excess sodium intake is a risk factor for developing NCDs such as hypertension and cardio-cerebrovascular disease, as well as alcohol use, smoking, and lack of physical activity [23]. Reducing sodium intake can contribute to reducing the prevalence and mortality caused by these diseases. Low-salt preference is known as an index reflecting individual's attitudes and reactions to salty taste [24, 25]. Therefore, low-salt preference influences an individual's sodium intake behaviour, such as the amount of salt that can be adjusted during the cooking process or at the table, the frequency of eating out and consumption of instant foods, and the amount of soup in the daily diet [26]. The significance of this study lies in the fact that it demonstrated the need to develop more effective strategies for the formation of low-salt dietary habits in major NCD patients by identifying the trend of low-salt preferences of major NCD patients using big data from the Republic of Korea.

In this study, the low-salt preference of patients with major NCD was found to be higher than that of the general population. From this finding, it can be inferred that NCD patients

were interested in a low-salt diet after being diagnosed with NCDs [27], or that they received nutrition education in healthcare facilities [28]. However, no significant differences in low-salt preference were observed between NCD patients and the general population, and the low salt preference rates gradually decreased after peaking in 2013. In Republic of Korea, the multi-component national programs to reduce salt intake has been steadily promoted since the 2010s [29]. In particular, in 2012, a national plan was established to reduce sodium intake. However, we estimated that this effect appears to have gradually weakened. This highlights the need to establish and implement more active interventions for a low-salt diet in NCD patients as well as the general population.

More specifically, among the items used to determine a low-salt preference, NCD patients showed higher rates for the item regarding the overall low-salt diet preference (usually eating bland or less salty food) when compared to the general population; however, the absolute value was lower than 30%. Sensitivity and familiarity to salty taste is affected by the concentration of sodium in foods that have been frequently exposed recently [30, 31], to form the habit of eating less salty food, efforts are needed to reduce the sodium content of the foods provided at home, school, workplace, and at other places regularly visited. In addition, efforts should be made to reduce the sodium content in convenience foods and delivery foods whose consumption continues to increase due to the coronavirus disease 2019 pandemic [32]. Government-led policies to reduce sodium intake through food service companies may be of great help in getting individuals to comply with a low-salt diet. However, given the lack of publicity for this policy and limited voluntary participation in the turnover-driven foodservice industry, stronger measures (e.g., a sodium tax) are required.

Another noteworthy finding of this study is that the rate of not adding salt and soy sauce at the table was higher than 70%. Since Korean food is mainly seasoned with salt or soy sauce during cooking, sodium intake can be significantly increased if it is added to the table. The high rate of a low-salt preference at the table is a desirable trend because, according to the health belief model [33], NCD patients recognise the severity of their diseases and make efforts to adhere to a low-salt diet, which is reflected in their avoidance of using salt and soy sauce. However, due to the low percentage of a low-salt preference (type III), it may be inferred that they do not add salt or soy sauce at the table because they usually eat sufficiently salted food. On the other hand, the rate of not dipping fried foods in soy sauce was lower than 50%. Fried foods are served with soy sauce because they are not strongly salted during cooking, and people tend to dip them in soy sauce. Therefore, to promote a low-salt diet, it is necessary to create a supportive environment such as educating cooks and foodservice professionals on recipes that use less salt or providing publicity about the importance of low-sodium diets for health.

It is also worth noting that greater regional variations were observed in low-salt preference among NCD patients compared to the general population. This gap may be attributable to the differences in disease prevalence, income level, and medical resources from one region to another, but may also be associated with sodium reduction policy programs run by local governments. Furthermore, it is necessary to pay attention to various characteristics related to salt intake, such as average temperature and diet, in order to explain these variations (S2 File). Just as there is variation in salt intake between countries [34], within a country, there may be differences in salt intake or salt preference from region to region. However, it is difficult to find studies examining which factors are most strongly associated with these variations. In the future, it would be meaningful to conduct studies to identify the causes of the regional differences in low-salt preference among NCD patients and to narrow the regional gap. The findings from these studies would ultimately contribute to resolving the regional disparities in healthy life expectancy [35].

This study has three limitations. First, only the variables included in the raw data were considered for the analysis because of the characteristics of the data source. In particular, the incidence of NCDs was identified by self-report, and the timing of diagnosis was unknown. In future research, it will be necessary to follow changes in dietary behaviours before and after diagnosis of the disease using other data sources such as panel data or patient cohort data. Second, although various policies regarding sodium intake reduction have been put in place in the Republic of Korea, their effects could not be examined in this study. To more effectively promote a low-sodium diet, it is necessary to conduct long-term follow-up studies that consider changes induced by such policies. Third, this study included 13 patients with NCDs, including hypertension and cardiovascular disease, which are closely associated with sodium intake. However, given that patients with cancer (e.g., stomach cancer) are also affected by a low-salt diet. The KCHS does not include a question asking whether participants have cancer. In future research, it is necessary to examine the health behaviours of cancer patients by using other data sources that include cancer patients.

## Conclusion

The results of this study highlight the need to improve the low-salt preference rate in patients with major NCDs as well as in the general population. Specifically, it is necessary to consider expanding various low-salt policies and programs such as education on a low-salt diet, salinity monitoring to create a low-salt environment, certification for low-sodium restaurants, and sodium tax. In these contexts, the low-salt preference rates among major NCD patients, as determined in this study, will serve as baseline values for evaluating the effectiveness of various low-salt policies and programs. These findings can also be used as reference values for setting project targets. Therefore, it is necessary to continuously monitor the low-salt preferences of patients with major NCDs and the general population.

## Supporting information

**S1 File. Regional variations in low-salt preference.**
(PDF)

**S2 File. Regional characteristics related to sodium intake.**
(PDF)

## Author Contributions

**Conceptualization:** Eun Young Choi, Young-Kwon Park, Minsu Ock.

**Data curation:** Young-Kwon Park, Minsu Ock.

**Formal analysis:** Young-Kwon Park, Minsu Ock.

**Methodology:** Eun Young Choi, Minsu Ock.

**Project administration:** Minsu Ock.

**Supervision:** Minsu Ock.

**Visualization:** Eun Young Choi, Young-Kwon Park.

**Writing – original draft:** Eun Young Choi, Minsu Ock.

**Writing – review & editing:** Eun Young Choi, Young-Kwon Park, Minsu Ock.

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
