## [Decision Letter · Decision Letter 0]

14 Aug 2022

PONE-D-22-18808Comparison of low-salt preference trends and regional variations between patients with major non-communicable diseases and the general populationPLOS ONE

Dear Dr. Ock,

Thank you for submitting your manuscript to PLOS ONE. After careful consideration, we feel that it has merit but does not fully meet PLOS ONE’s publication criteria as it currently stands. Therefore, we invite you to submit a revised version of the manuscript that addresses the points raised during the review process.

Two reviewers well assessed this manuscript.However, there are still corrections existed in the present form of the manuscript.

Read the suggestions carefully and respond them appropriately.

We look forward to receiving your revised manuscript.

Kind regards,

Masaki Mogi

Academic Editor

PLOS ONE

Journal Requirements:

"This work was supported by the Research Program funded by the Korea Disease Control and Prevention Agency. The funders had no role in study design, data collection and analysis, decision to publish, or preparation of the manuscript."

Reviewers' comments:

Reviewer's Responses to Questions

**Comments to the Author**

1. Is the manuscript technically sound, and do the data support the conclusions?

Reviewer #1: Yes

Reviewer #2: Yes

2. Has the statistical analysis been performed appropriately and rigorously? 

Reviewer #1: Yes

Reviewer #2: I Don't Know

3. Have the authors made all data underlying the findings in their manuscript fully available?

Reviewer #1: Yes

Reviewer #2: Yes

4. Is the manuscript presented in an intelligible fashion and written in standard English?

Reviewer #1: Yes

Reviewer #2: Yes

5. Review Comments to the Author

Reviewer #1: Dear Authors,

The article provides a good insight into the trends in low-salt preference and regional differences in Korea among patients with the main non-communicable diseases.

For a better understanding of the work, I have some suggestions:

Line 118.

Can the authors explain the background or give a reference why the 13 NCD listed are the main ones?

Line 208:

To the reader who does not know Korea well, the names of the regions do not give useful information.

Would it make sense to describe each region (in few words) background that could influence salt consumption?

For example,

-a coastal region, traditionally lot of fish are consumed,…

-an inland region, …

-rural region,... intensive urban region,..

-a region where traditionally few meat is consumed...

Lines 234 235:

The low-salt preference rates of patients with different major NCDs and the general population was gradually decreasing after peaking mainly in 2013

The trend is evident from figures 2,3,4,5, can you comment the reasons for the trend?

Line 305:

In the title of the article you "regional variations" are "highlighted

Can you briefly comment the reasons for the differences in patient behavior between regions?

Best regards.

Reviewer #2: This study deals with an important issue within Korea as well as worldwide. The study uses the results of a large survey to analyze trends over time in the differences in low salt preference between major NCD patients and the general population, and offers suggestions for future environmental improvement and education. However, there are some concerns in this paper.

1. Abstract: Result: The authors specifically describe the low salt preference rate. Since the survey years for this study are 2008-2019, Please indicate the year of the survey.

2. Introduction: 87: The survey years are 2008-2017, but is it 2008-2019?

3. Materials and methods: What is the size of The Korea Community Health Survey? Please provide sample size information for each survey year.

4.Materials and methods: Participants: The authors do not include cancer as Major NCD Patients, although they mention it in the study limitations. Why not include cancer in this analysis?

In addition, since the results and discussion focus on hypertension, diabetes, and dyslipidemia, it would be better to exclude other diseases. I think it would be better to have the data of Fig2-Fug5 to grasp the overall trend of NCD, but I feel it is difficult to confirm the data.

5.Results:178-179:　“In the general population, it peaked in 2013 at 74.4% (APC: -11.788, p-value: <0.001)”. Isn't the APC value 11.788?

6.Results:184-185：Other NCD results are not mentioned in the results that follow. If the relevant data set is not always available, then it may be possible to remove it.

7. Results:194：Please list the ACP values for 2013.

8. Results:201：Please correct the p-value.

9. Discussion: The low salt preference rate and the adherence rate of not adding salt or soy sauce at the table have changed since 2013, but the reasons for this have not been mentioned. Is there any possible reason for this change, such as a change in the survey methodology, the selection method of the target population, or a change in government policy policy regarding salt reduction?

10. Discussion：260-261：“the reaction to a salty taste is influenced by the sodium concentration in the food”. It would be helpful to the reader's understanding if you could describe the specific impacts while presenting the reports of the cited references.

11. Discussion：260-261：Please describe in detail the contents of your education to cooks and foodservice professionals.

12. Disccussion：287-288：Please cite any reports showing that low-salt preference is related to temperature.

6. PLOS authors have the option to publish the peer review history of their article (what does this mean?). If published, this will include your full peer review and any attached files.

Reviewer #1: No

Reviewer #2: No

---

## [Author Response · Author response to Decision Letter 0]

23 Sep 2022

Reviewers' comments:

Reviewer's Responses to Questions

Comments to the Author

1. Is the manuscript technically sound, and do the data support the conclusions?

Reviewer #1: Yes

Reviewer #2: Yes

2. Has the statistical analysis been performed appropriately and rigorously?

Reviewer #1: Yes

Reviewer #2: I Don't Know

3. Have the authors made all data underlying the findings in their manuscript fully available?

Reviewer #1: Yes

Reviewer #2: Yes

4. Is the manuscript presented in an intelligible fashion and written in standard English?

Reviewer #1: Yes

Reviewer #2: Yes

5. Review Comments to the Author

Reviewer #1: Dear Authors,

The article provides a good insight into the trends in low-salt preference and regional differences in Korea among patients with the main non-communicable diseases.

For a better understanding of the work, I have some suggestions:

Response: We really appreciate the careful and thorough review of this manuscript and for the thoughtful comments and constructive suggestions, which help to improve the quality of the manuscript. Our responses are as follows.

Line 118.

Can the authors explain the background or give a reference why the 13 NCD listed are the main ones? 

Response: Thirteen major types of NCD patients (hypertension, diabetes, dyslipidaemia, stroke, myocardial infarction, angina, arthritis, osteoporosis, asthma, allergic rhinitis, atopic dermatitis, cataract, and depression) were defined as participants who responded clinically diagnosed from physicians in the KCHS. We have revised the sentences (line 117-120).

Line 208:

To the reader who does not know Korea well, the names of the regions do not give useful information.

Would it make sense to describe each region (in few words) background that could influence salt consumption?

For example,

-a coastal region, traditionally lot of fish are consumed,…

-an inland region, …

-rural region,... intensive urban region,..

-a region where traditionally few meat is consumed...

Response: Thanks for the constructive suggestion. As suggested, we have added regional characteristics of Republic of Korea, including sodium intake, as a Supplemental file. Sentences that further research on regional variation are needed have been added to the Discussion section (Line 292-297). However, we ask for your understanding that we cannot cover various regional characteristics due to data limitations.

Lines 234 235:

The low-salt preference rates of patients with different major NCDs and the general population was gradually decreasing after peaking mainly in 2013

The trend is evident from figures 2,3,4,5, can you comment the reasons for the trend?

Response: In Republic of Korea, the multicomponent national programs to reduce salt intake has been steadily promoted since the 2010s. In particular, in 2012, a national plan was established to reduce sodium intake. However, we estimated that this effect appears to have gradually weakened. These contents were further reinforced in the Discussion section (Line 256-259).

Line 305:

In the title of the article you "regional variations" are "highlighted

Can you briefly comment the reasons for the differences in patient behavior between regions?

Response: As mentioned in the previous response, the need for additional studies on regional variation was emphasized in the Discussion section (Line 292-297).

Reviewer #2: This study deals with an important issue within Korea as well as worldwide. The study uses the results of a large survey to analyze trends over time in the differences in low salt preference between major NCD patients and the general population, and offers suggestions for future environmental improvement and education. However, there are some concerns in this paper.

Response: We really appreciate the careful and thorough review of this manuscript and for the thoughtful comments and constructive suggestions, which help to improve the quality of the manuscript. Our responses are as follows.

1. Abstract: Result: The authors specifically describe the low salt preference rate. Since the survey years for this study are 2008-2019, Please indicate the year of the survey.

Response: In the Materials and methods section, we have already mentioned that the Korea Community Health Survey is conducted annually (Line 110). The Korea Community Health Survey results for each year were used for analysis.

2. Introduction: 87: The survey years are 2008-2017, but is it 2008-2019?

Response: In the previous study (Reference number 10), the results of the Korea Community Health Survey from 2008 to 2017 were analyzed to analyze health behaviors, such as smoking and drinking, among major NCD patients. This content is a description of prior study.

3. Materials and methods: What is the size of The Korea Community Health Survey? Please provide sample size information for each survey year.

Response: As suggested, we added information about sample size from the Korea Community Health Survey to the Materials and methods section (Line 113-114).

4.Materials and methods: Participants: The authors do not include cancer as Major NCD Patients, although they mention it in the study limitations. Why not include cancer in this analysis?

Response: Unfortunately, the Korea Community Health Survey does not include a question asking whether participants have cancer. We emphasize this point further in the part on limitations in the Discussion section (Line 312-314).

In addition, since the results and discussion focus on hypertension, diabetes, and dyslipidemia, it would be better to exclude other diseases. I think it would be better to have the data of Fig2-Fug5 to grasp the overall trend of NCD, but I feel it is difficult to confirm the data.

Response: As you know, among various NCDs, hypertension, diabetes, and dyslipidemia are important diseases that act as risk factors for other diseases. Also, these diseases have a higher prevalence than other diseases. So, Figure 1 focused on hypertension, diabetes, and dyslipidemia. In addition, a Supplemental file 1 is attached so that readers can check the specific values for each disease.

5.Results:178-179:　“In the general population, it peaked in 2013 at 74.4% (APC: -11.788, p-value: <0.001)”. Isn't the APC value 11.788?

Response: As you pointed out, this is a typo. We have corrected that value (Line 182).

6.Results:184-185：Other NCD results are not mentioned in the results that follow. If the relevant data set is not always available, then it may be possible to remove it.

Response: As suggested, we have deleted that sentence.

7. Results:194：Please list the ACP values for 2013.

Response: It has a single APC with respect to its content. That is, the APC is -0.152.

8. Results:201：Please correct the p-value.

Response: We have modified the p-value (Line 202).

9. Discussion: The low salt preference rate and the adherence rate of not adding salt or soy sauce at the table have changed since 2013, but the reasons for this have not been mentioned. Is there any possible reason for this change, such as a change in the survey methodology, the selection method of the target population, or a change in government policy policy regarding salt reduction?

Response: In Republic of Korea, the multicomponent national programs to reduce salt intake has been steadily promoted since the 2010s. In particular, in 2012, a national plan was established to reduce sodium intake. However, we estimated that this effect appears to have gradually weakened. These contents were further reinforced in the Discussion section (Line 256-261).

10. Discussion：260-261：“the reaction to a salty taste is influenced by the sodium concentration in the food”. It would be helpful to the reader's understanding if you could describe the specific impacts while presenting the reports of the cited references.

Response: As suggested, we revised the sentence (line 265-266).

11. Discussion：260-261：Please describe in detail the contents of your education to cooks and foodservice professionals.

Response: The content of the sentence you pointed out has been modified to be more specific (Line 286-287).

12. Disccussion：287-288：Please cite any reports showing that low-salt preference is related to temperature.

Response: We have corrected the sentence and added a sentence that additional research on regional variation is needed (Line 292-297).

6. PLOS authors have the option to publish the peer review history of their article (what does this mean?). If published, this will include your full peer review and any attached files.

Do you want your identity to be public for this peer review? For information about this choice, including consent withdrawal, please see our Privacy Policy.

Reviewer #1: No

Reviewer #2: No

---

## [Decision Letter · Decision Letter 1]

12 Oct 2022

Comparison of low-salt preference trends and regional variations between patients with major non-communicable diseases and the general population

PONE-D-22-18808R1

Dear Dr. Ock,

We’re pleased to inform you that your manuscript has been judged scientifically suitable for publication and will be formally accepted for publication once it meets all outstanding technical requirements.

Kind regards,

Masaki Mogi

Academic Editor

PLOS ONE

Additional Editor Comments (optional):

The authors well responded to the Reviewers' comments. 

Reviewers' comments:

Reviewer's Responses to Questions

**Comments to the Author**

1. If the authors have adequately addressed your comments raised in a previous round of review and you feel that this manuscript is now acceptable for publication, you may indicate that here to bypass the “Comments to the Author” section, enter your conflict of interest statement in the “Confidential to Editor” section, and submit your "Accept" recommendation.

Reviewer #1: All comments have been addressed

Reviewer #2: All comments have been addressed

2. Is the manuscript technically sound, and do the data support the conclusions?

Reviewer #1: Yes

Reviewer #2: Yes

3. Has the statistical analysis been performed appropriately and rigorously? 

Reviewer #1: Yes

Reviewer #2: Yes

4. Have the authors made all data underlying the findings in their manuscript fully available?

Reviewer #1: Yes

Reviewer #2: Yes

5. Is the manuscript presented in an intelligible fashion and written in standard English?

Reviewer #1: Yes

Reviewer #2: Yes

6. Review Comments to the Author

Reviewer #1: Dear Authors,

Thanks for the comments and inclusion of reccomendations.

I wish you successful further research work.

Best regards

Reviewer #2: (No Response)

7. PLOS authors have the option to publish the peer review history of their article (what does this mean?). If published, this will include your full peer review and any attached files.

Reviewer #1: No

Reviewer #2: No

---

## [Editor Report · Acceptance letter]

17 Oct 2022

PONE-D-22-18808R1 

Comparison of low-salt preference trends and regional variations between patients with major non-communicable diseases and the general population 

Dear Dr. Ock:

I'm pleased to inform you that your manuscript has been deemed suitable for publication in PLOS ONE. Congratulations! Your manuscript is now with our production department. 

Kind regards, 

on behalf of

Dr. Masaki Mogi 

Academic Editor

PLOS ONE